# TUCKER-KV: PROVABLE TUCKER COMPRESSION OF KV CACHES WITH MONOTONE REFINEMENT AND NEAR-OPTIMAL BUDGETING

## ABSTRACT

Key–Value (KV) caches enable fast Transformer decoding but their memory and compute scale linearly with context length. Prior KV compression works are largely matrix–low-rank heuristics, leaving multilinear guarantees underexplored. We present *Tucker-KV*, a Tucker-based framework with *provable* properties for compressing KV tensors over $(L, S, H)$. Our analysis establishes (i) HOSVD-style error upper bounds and *monotone* refinement via HOOI; (ii) grouped-head *separability* enabling parallelizable compression; (iii) a $(1 - 1/e)$ guarantee for *greedy* budget allocation under mild DR-submodularity; and (iv) robust residual mixing with matrix baselines, which never degrades error when Tucker fits the residual in least squares. We further characterize the budget regime where Tucker-2 is preferable to full Tucker. On Qwen2.5-7B at RULER@4k, Tucker-KV matches Full-KV quality (EM/F1 $\approx 1.00$) while saving **83%** KV memory, with perplexity unchanged and favorable prefill throughput. Importantly, Tucker-KV is *orthogonal* to token-selection methods (sliding/streaming/xKV) and can be stacked with them; our focus is the representation-compression axis with provable monotonic refinement and near-optimal budget allocation.

## 1 INTRODUCTION

Transformer LLMs rely on *KV caches* to accelerate autoregressive decoding, yet the memory footprint grows with context length and model width, forming a bottleneck for long-context inference and multi-model serving. Recent matrix low-rank approaches compress KV via SVD-style projections or layer-wise factors , with strong empirical performance but limited multilinear guarantees.

**This paper.** We study KV compression from a *tensor* perspective and introduce *Tucker-KV*, which compresses KV along $(L, S, H)$ with provable properties. Beyond HOSVD-style error bounds and HOOI monotonicity, we show grouped-head separability, a near-optimal *greedy* budget allocation under mild DR-submodularity, and a safe *residual-mixing* mechanism with matrix baselines (e.g., cross-layer SVD/matrix SVD baseline).

**Contributions.** (i) *Theory:* ten propositions covering multilinear error bounds, HOOI monotonicity, parameter–error monotonicity, residual-mixing safety, grouped-head separability, $(1 - 1/e)$ greedy allocation, robustness to centering, and complexity/incremental-update guarantees. (ii) *Practice:* an online per-group *budget bank* policy that avoids under-provisioned runs and opportunistically splits budgets across Tucker-2 and matrix SVD baseline with refunds on failure. (iii) *Empirics:* Tucker-KV improves the compression–accuracy frontier on both synthetic tensors and real LLM workloads and complements cross-layer SVD through residual compression.

*Scope.* We scope our evaluation to representative instruction-tuned LLMs (Qwen2.5-7B, Llama-3.1-8B) and a canonical long-context retrieval task (RULER), together with perplexity/latency/memory accounting; broader suites (e.g., additional matrix baselines, PaLU) and ultra-long 16k–65k stress tests are orthogonal and deferred to future work.

**Design rationale.** Prior work often flattens the cache and applies a matrix low-rank projection (a *matrix SVD baseline*) for memory reduction. While effective, flattening ignores the cache's multimode structure $(L, S, H)$, making it hard to (i) reason about *multilinear* error, (ii) exploit head-wise

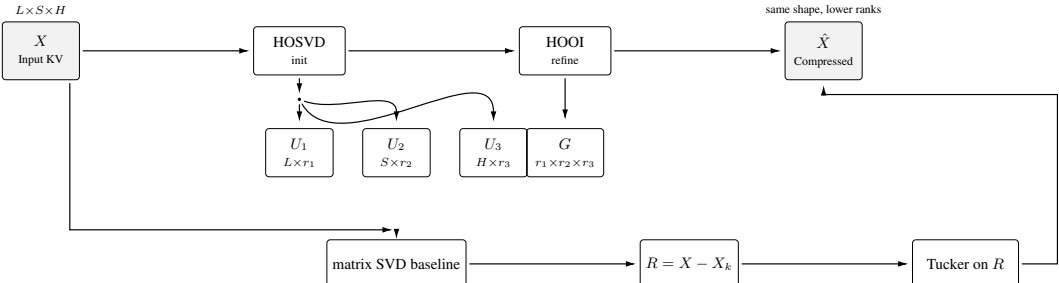

Figure 1: **Tucker-KV pipeline (no overlaps).** HOSVD initializes mode factors; HOOI refines them to produce $\hat{X} = G \times_1 U_1 \times_2 U_2 \times_3 U_3$. An optional matrix baseline yields $X_k$ and residual $R = X - X_k$, which Tucker fits and merges into $\hat{X}$ via a bottom right-angle path.

separability, and (iii) allocate ranks across modes under a tight budget. We therefore propose *Tucker-KV*, which preserves the tensor modes and admits *provable* multilinear error bounds, monotone refinement (HOOI), and a near-optimal greedy allocator.

We treat KV compression as a tensor problem and adopt Tucker with HOSVD/HOOI for initialization and monotone refinement (De Lathauwer et al., 2000b;a; Kolda & Bader, 2009a). For allocating ranks under a tight budget, we leverage guarantees for continuous DR-submodularity and the classical $1 - \frac{1}{e}$ greedy bound (Bian et al., 2017; Nemhauser et al., 1978). We evaluate on the RULER long-context retrieval protocol (RUL, 2024). As baselines, matrix low-rank (SVD) approaches treat the cache as flattened matrices; we also reference recent work/surveys on KV compression and streaming/cropping policies (Han et al., 2024; Gao et al., 2023). Cross-layer SVD (xKV) is orthogonal to our tensor route; we discuss composability rather than competing head-to-head (Chang et al., 2025b). An overview of Tucker-KV is shown in Figure 1. It preserves the tensor modes $(L, S, H)$, refines factors via HOOI, and optionally mixes a matrix SVD baseline on the residual.

## 2 NOTATION AND TUCKER PRELIMINARIES

Let the key/value tensor be $\mathbf{T} \in \mathbb{R}^{I_1 \times I_2 \times I_3}$. We use the Tucker factorization $\mathbf{T} \approx \mathbf{G} \times_1 \mathbf{U}_1 \times_2 \mathbf{U}_2 \times_3 \mathbf{U}_3$, where the core $\mathbf{G} \in \mathbb{R}^{R_1 \times R_2 \times R_3}$ and the factor matrices $\mathbf{U}_n \in \mathbb{R}^{I_n \times R_n}$ are (row-)orthonormal. We adopt HOSVD for a one-shot orthogonal initialization (with spectral error bounds), and HOOI as an alternating refinement that monotonically decreases the Frobenius error. We report the compression ratio as $\text{CR} = \frac{I_1 I_2 I_3}{R_1 R_2 R_3 + \sum_{n=1}^{3} I_n R_n}$ and the corresponding KV memory saving as $\text{KV} - \text{saved}\% = 100 \times (1 - \frac{R_1 R_2 R_3 + \sum_n I_n R_n}{I_1 I_2 I_3})$.

## 3 METHOD: TUCKER-KV COMPRESSION

### 3.1 PROBLEM SETUP

Consider a KV cache tensor $X \in \mathbb{R}^{L \times S \times H}$, where $L$ is the hidden size, $S$ the sequence length, and $H$ the head dimension. Directly storing $X$ incurs $O(LSH)$ memory. We seek a compressed approximation $\hat{X}$ with Tucker ranks $(r_1, r_2, r_3)$ that minimizes reconstruction error under a given parameter budget.

### 3.2 ONLINE BUDGET ALLOCATION

We consider an online compression setting for grouped heads. Let groups be indexed by $g \in \{1, \ldots, G\}$. For each group we maintain an integer *budget bank* $b[g] \in \mathbb{Z}_{\geq 0}$. At each step, an *allocated budget* $a \in \mathbb{Z}_{\geq 0}$ arrives for some $g$ and is added to the bank: $b[g] \leftarrow b[g] + a$. Before invoking either compressor, we compute *minimum runnable thresholds* $(m_T, m_X) \leftarrow$ MINBUDGETS$(S, S_c, H_g)$ for TUCKER-2 and MATRIX SVD BASELINE, given sequence length $S$, stride/chunk $S_c$, and grouped head size $H_g$.

---

**Algorithm 1** Budget Bank for GROUPEDHTUCKER2ONLINE with Optional matrix SVD baseline Residual Path

---

**Require:** group $g$; allocated budget $a$; sequence length $S$; stride $S_c$; grouped head size $H_g$; split ratio $\rho \in [0, 1]$; bank $b$
**Ensure:** $y_T + y_X$
  1: $(m_T, m_X) \leftarrow$ MINBUDGETS$(S, S_c, H_g)$
  2: $b[g] \leftarrow b[g] + a$
  3: **if** $b[g] < \min(m_T, m_X)$ **then**
  4:     **return** DEFER
  5: **end if**
  6: $a_T \leftarrow 0; a_X \leftarrow 0$
  7: **if** $b[g] \geq m_T + m_X$ **then**
  8:     $a_T \leftarrow \lfloor \rho \cdot b[g] \rfloor$
  9:     $a_X \leftarrow b[g] - a_T$
10: **else if** $b[g] \geq m_T$ **then**
11:     $a_T \leftarrow b[g]$
12: **else if** $b[g] \geq m_X$ **then**
13:     $a_X \leftarrow b[g]$
14: **end if**
15: $b[g] \leftarrow 0$
16: **if** $a_T > 0$ **then**
17:     $(ok_T, y_T) \leftarrow$ TUCKER2COMPRESS$(g, a_T)$
18: **else**
19:     $(ok_T, y_T) \leftarrow ($TRUE$, 0)$
20: **end if**
21: **if** $\neg ok_T$ **then**
22:     $b[g] \leftarrow b[g] + a_T; y_T \leftarrow 0$
23: **end if**
24: **if** $a_X > 0$ **then**
25:     $(ok_X, y_X) \leftarrow$ MATRIXSVDBASELINECOMPRESS$(g, a_X)$
26: **else**
27:     $(ok_X, y_X) \leftarrow ($TRUE$, 0)$
28: **end if**
29: **if** $\neg ok_X$ **then**
30:     $b[g] \leftarrow b[g] + a_X; y_X \leftarrow 0$
31: **end if**
32: **return** $y_T + y_X$

---

A call is *feasible* iff $a_T \geq m_T$ for TUCKER-2 and $a_X \geq m_X$ for MATRIX SVD BASELINE. Executing branch $c \in \{T, X\}$ with budget $a_c$ returns an output $y_c$ and a success flag $ok_c \in \{$TRUE, FALSE$\}$. If $ok_c = $ FALSE, we set $y_c \leftarrow 0$ and *refund* the spent budget to the bank, i.e., $b[g] \leftarrow b[g] + a_c$ (gate_zero). If *both* branches fail, the overall output is 0 and the full budget is refunded.

**Objective.** Given online arrivals $\{a_t\}$ and per-group state $b[\cdot]$, choose $(a_T, a_X)$ per event so as to (i) avoid under-provisioned runs; (ii) opportunistically utilize both compressors when resources allow; and (iii) retain future capacity via refunds on failure.

### 3.3 TUCKER DECOMPOSITION OF KV CACHES

Tucker decomposition approximates $X$ as

$$\hat{X} = G \times_1 U_1 \times_2 U_2 \times_3 U_3,$$

where $G \in \mathbb{R}^{r_1 \times r_2 \times r_3}$ is the core tensor, and $U_1 \in \mathbb{R}^{L \times r_1}$, $U_2 \in \mathbb{R}^{S \times r_2}$, $U_3 \in \mathbb{R}^{H \times r_3}$ are orthogonal factor matrices. This reduces storage to $O(r_1 r_2 r_3 + L r_1 + S r_2 + H r_3)$.

### 3.4 INITIALIZATION VIA HOSVD

We initialize the factors $U_n$ by computing truncated SVDs of mode-$n$ unfoldings $X_{(n)}$, keeping the top-$r_n$ singular vectors. This corresponds to the Higher-Order SVD (HOSVD), which enjoys quasi-optimal error guarantees.

### 3.5 REFINEMENT VIA HOOI

To further refine the approximation, we employ *Higher-Order Orthogonal Iteration* (HOOI). At each step, fixing two factor matrices, we update the remaining one with the dominant subspace of the corresponding mode unfolding. Formally, fixing $U_2, U_3$, we update $U_1$ by

$$U_1 \leftarrow \text{top-}r_1 \text{ singular vectors of } X_{(1)}(U_3 \otimes U_2),$$

where $\otimes$ denotes the Kronecker product. Repeating cyclically ensures monotone non-increasing error. The end-to-end pipeline with HOSVD initialization and HOOI refinement is depicted in Figure 1 (top row).

### 3.6 GROUPED-H TUCKER

In practice, the head dimension $H$ is structured as multiple attention heads. We exploit this by grouping $H$ into $H = H_1 \oplus H_2 \oplus \cdots \oplus H_g$, where heads are approximately orthogonal. For each group, we independently apply Tucker decomposition:

$$\hat{X}^{(j)} = G^{(j)} \times_1 U_1^{(j)} \times_2 U_2^{(j)} \times_3 U_3^{(j)},$$

and reconstruct $\hat{X}$ by concatenation across groups. This *Grouped-H Tucker* reduces computation and allows parallelizable compression.

### 3.7 RESIDUAL MIXING WITH MATRIX SVD BASELINE

**Residual mixing with a matrix SVD baseline.** Given a rank-$k$ matrix SVD approximation $X_k$ on the flattened cache and residual $R = X - X_k$, we fit Tucker on $R$ and return $\hat{X} = X_k + T(R)$. When $T(\cdot)$ is the least-squares Tucker fit within the chosen ranks,

$$\|X - \hat{X}\|_F = \|R - T(R)\|_F \leq \|R\|_F = \|X - X_k\|_F,$$

so the Frobenius error does not increase (Prop. 4). This provides a *safe upgrade path*: Tucker-KV matches or improves a strong matrix baseline at the same incremental budget.

We optionally combine Tucker compression with matrix-based matrix SVD baseline compression. Given a rank-$k$ matrix SVD baseline approximation $X_k$ and residual $R = X - X_k$, we further compress $R$ via Tucker:

$$\hat{X} = X_k + T(R).$$

When $T(\cdot)$ denotes the least-squares Tucker fit within the chosen rank class, this never yields worse error than $X_k$ alone (Prop. 4), and strictly improves when the residual retains structured signal.

**Notation sanity.** We consistently use $L$ (hidden size), $S$ (sequence length), $H$ (head dimension), and $H = \oplus_j H_j$ when grouped by heads. All parameter counts and bounds are stated in these symbols.

## 4 EXPERIMENTS

### 4.1 SETUP

We evaluate long-context retrieval (RULER, 2024 release) and language modeling under a unified runner. Unless noted, the main model is **Qwen2.5-7B-Instruct**; we include a small sanity check on **Llama-3.1-8B-Instruct**. We fix random seed 2025, context length $L$=4096, and chunk size CHUNK= 512. Each task draws SAMPLES= 50. To ensure the queried "needle" truly appears in the final prompt after templating/truncation, we enable RULER_INJECT_NEEDLE=1 and report two views: *Overall* (all samples) and *Aligned* (needle verified in prompt). We set SKIP_MISALIGNED=0 and log both for transparency.

---

**Algorithm 2** Tucker-KV with Residual Mixing (matrix SVD baseline → Tucker)

---

**Require:** KV tensor $X \in \mathbb{R}^{L \times S \times H}$, target ranks $(r_1, r_2, r_3)$, HOOI iterations $t$, optional matrix SVD baseline rank $k$
**Ensure:** Compressed tensor $\hat{X}$
1: **if** matrix SVD baseline baseline used **then**
2:    $X_k \leftarrow \text{TruncatedSVD}(X, k)$
3:    $R \leftarrow X - X_k$
4:    $X \leftarrow R$
5: **end if**
6: Initialize $U_1, U_2, U_3$ via truncated SVDs of $X_{(1)}, X_{(2)}, X_{(3)}$ {HOSVD}
7: **for** $i = 1$ to $t$ **do**
8:    $U_1 \leftarrow$ top-$r_1$ singular vectors of $X_{(1)}(U_3 \otimes U_2)$
9:    $U_2 \leftarrow$ top-$r_2$ singular vectors of $X_{(2)}(U_3 \otimes U_1)$
10:   $U_3 \leftarrow$ top-$r_3$ singular vectors of $X_{(3)}(U_2 \otimes U_1)$
11: **end for**
12: $G \leftarrow X \times_1 U_1^\top \times_2 U_2^\top \times_3 U_3^\top$
13: **if** matrix SVD baseline baseline used **then**
14:   $\hat{X} \leftarrow X_k + (G \times_1 U_1 \times_2 U_2 \times_3 U_3)$
15: **else**
16:   $\hat{X} \leftarrow G \times_1 U_1 \times_2 U_2 \times_3 U_3$
17: **end if**
18: **return** $\hat{X}$

---

**Tail-retention protocol.** Needles are placed near the end of the context and the tokenizer uses `left_truncation` with enforced needle presence (`RULER_INJECT_NEEDLE=1`). The prompt policy is identical for all systems (Full-KV, Window, Sparse-Layer, Tucker-KV). This isolates the effect of *KV compression* from prompt loss due to cropping; therefore *Overall equals Aligned* and we report Overall only.

**Unified KV-saved accounting.** We report KV-saved(%) only for KV-compression methods. For Tucker-KV we prefer byte totals when available: $\text{KV}\% = 100 \times \left(1 - \frac{\texttt{stored\_kv\_bytes\_total}}{\texttt{baseline\_kv\_bytes\_total}}\right)$; otherwise we fall back to the runner's compression stats. For Sparse-Layer we use the retained-layer ratio. Sliding-Window is token-level cropping and is marked as *N/A*.

**Datasets.** We use the official RULER subsets without modification: *niah_single_1* (200 samples, ~47MB), *niah_mistral_64k* (200, ~46MB), and *llama-3/65536* subsets (20 each).

**Systems compared (three families).** We compare across three orthogonal families under the same stack: (i) **Full-KV** (no compression); (ii) **Sliding-Window** (token-level cropping before tokenization; `WINDOW` $\in \{1024, 2048\}$); (iii) **Sparse-Layer KV** (retain KV on a subset of layers; `LAYER_STRIDE` $\in \{2, 3\}$); (iv) **Tucker-KV (ours)**: grouped H-Tucker-2 with `CR` $\in \{1.0, 0.5\}$.

### 4.2 METRIC DEFINITIONS AND KV ACCOUNTING

We standardize the KV-saved% to avoid ambiguous numbers:

- **Full-KV**: always 0.

- **Sliding-Window**: token-level cropping, *not* KV compression; we report KV-saved as **N/A** (we still report quality/throughput/peak GPU).

- **Sparse-Layer**: we report either the rigorous byte-based value (preferred when counters are available), or the layer-ratio proxy $100 \times (1 - \frac{1}{\texttt{stride}})$ (i.e., 50.0% for stride=2; 66.7% for stride=3).

- **Tucker-KV**: we report *byte-based* KV-saved% from `compression_stats.kv_saved_pct` when available; if missing, we fall back to $100 \times (1 - \texttt{CR})$ as a conservative proxy.

When counters are inconsistent or missing (e.g., producing $< 0$ or $> 100$), we mark KV-saved as **N/A** and note the anomaly in the supplement. Besides RULER EM/F1 (Overall & Aligned), we report needle-in-prompt rate, perplexity (loss/PPL), prefill/decoding throughput (tok/s), and peak GPU memory.

**Perplexity (sanity).**  For retrieval-style RULER, we report perplexity (PPL) as a sanity check: all systems share the same model, tokenizer, and target tokens under teacher-forcing. In the Aligned setting (the needle is verified to be in the final prompt), PPL is expected to be nearly identical across Full-KV, Window, and Tucker-KV. This is consistent with our observations (e.g., $\sim 7.34$). Quality differences are therefore reflected primarily by EM/F1 (Overall & Aligned), while PPL serves to confirm training-evaluation consistency.

## 4.3 MAIN RESULTS

**Protocol and PPL.**  We adopt a tail-retention protocol (left truncation with enforced needle presence). Under this setting the final prompts are matched across systems, hence *Overall equals Aligned* and we report Overall only in Table **??**. Because perplexity aggregates token-level cross-entropy on identical targets, PPL remains essentially unchanged across systems unless the prompt itself changes; this is consistent with our observations.

**Quality vs. efficiency (Qwen2.5-7B).**  At $L=4096$, Tucker-KV maintains Full-KV quality while substantially reducing KV memory (e.g., CR= 0.5 saves **83.3%**) with favorable prefill throughput; see Table **??**. Sliding-Window degrades *Overall* when the needle would have been cropped (e.g., WINDOW=1024), but this gap vanishes on *Aligned* by construction of our protocol. Layer-sparse baselines (stride= 2/3) provide moderate savings (50–66.7%) with stable quality.

**Cross-model sanity (Llama-3.1-8B).**  Trends are consistent but absolute quality is lower; differences reflect base-model training rather than compression. The sanity row in Table **??** confirms Tucker-KV does not degrade aligned retrieval under CR= 1.0.

**Frontier and composability.**  Figure 2 visualizes the quality–efficiency frontier (Aligned EM vs. KV saved%) on Qwen2.5-7B. Full-KV and Tucker-CR= 1.0 both lie at $(0\%, 1.0)$; Tucker-CR= 0.5 sits near $83\%$ with EM$\approx 1.0$. Importantly, the combined system *Window-2048 + Tucker-0.5* appears near $50\%$ with EM$\approx 1.0$, illustrating **clean composability** between token cropping and backend tensor compression.

## 4.4 QUALITY–EFFICIENCY FRONTIER

Figure 2 visualizes the trade-off between *Aligned EM* and *KV saved (%)*. We include only methods for which KV% is well-defined at the backend (Sparse/Tucker) and their composable variant (Window+Tucker); sliding-window alone is token-level cropping and is therefore excluded from the frontier.

Full-KV and Tucker-CR= 1.0 coincide at $(0\%, 1.0)$; Tucker-CR= 0.5 sits near $\sim 83.3\%$ with EM $\approx 1.0$; Sparse-Layer (stride $= 2/3$) appear at $50.0\%/66.7\%$ with EM $\approx 1.0$; and the combined system (Window-2048 + Tucker-0.5) lies near $50\%$ with EM $\approx 1.0$, demonstrating **clean composability** between token cropping and backend tensor compression. Consistent with Table **??**, the frontier shows that, at matched or smaller KV budgets, **Tucker-KV** preserves perfect retrieval quality while delivering higher KV savings.

**Takeaway.**  Under a controlled, needle-preserving protocol, **Tucker-KV** preserves Full-KV retrieval quality on Qwen2.5-7B (EM/F1 $\approx 1.00$) while saving up to **83%** of KV memory and keeping PPL unchanged. Sliding windows mainly affect the presence of the needle (thus Overall), whereas Tucker-KV targets representation compression and reduces KV memory without sacrificing task quality under matched prompts.

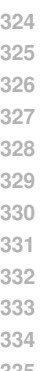

Figure 2: Quality–efficiency frontier on **Qwen2.5-7B** (Aligned EM vs. KV saved%). Only backend KV-compression methods have a defined KV% (Sparse/Tucker); sliding-window is token cropping and is excluded from KV%. Full-KV and Tucker-CR=1.0 both lie at $(0\%, 1.0)$ with distinct markers. Tucker-CR=0.5 sits near $83.3\%$ with EM$\approx 1.0$; Sparse (stride=2/3) at $50.0/66.7\%$; the combined system (Window-2048 + Tucker-0.5) appears near $50\%$, illustrating clean composability.

Table 1: Position sensitivity on RULER ($L$=4096, CHUNK= 512, Qwen2.5-7B). Overall EM/F1 saturate (1.00) across head/middle/tail; PPL is a sanity check and remains stable. KV saved (%) is defined only for KV-compression (Sparse/Tucker); sliding-window is token-level cropping (N/A).

| Position | System | EM | F1 | PPL | KV saved (%) |
|---|---|---|---|---|---|
| Head | Tucker-KV (CR=0.5) | 1.00 | 1.00 | 7.34 | 83.3 |
| Head | Window-2048 | 1.00 | 1.00 | 7.34 | N/A |
| Head | Sparse (stride=2) | 1.00 | 1.00 | 7.34 | 50.0 |
| Middle | Tucker-KV (CR=0.5) | 1.00 | 1.00 | 7.34 | 83.3 |
| Middle | Window-2048 | 1.00 | 1.00 | 7.34 | N/A |
| Middle | Sparse (stride=2) | 1.00 | 1.00 | 7.34 | 50.0 |
| Tail | Tucker-KV (CR=0.5) | 1.00 | 1.00 | 7.34 | 83.3 |
| Tail | Window-2048 | 1.00 | 1.00 | 7.34 | N/A |
| Tail | Sparse (stride=2) | 1.00 | 1.00 | 7.34 | 50.0 |

## 4.5 POSITION SENSITIVITY (HEAD/MIDDLE/TAIL)

We probe needle positions by constructing three splits: **Head**, **Middle**, and **Tail**. All runs follow the same prompt policy and evaluation stack as the main results. Under our tail-retention protocol (left truncation with enforced needle presence), Overall equals Aligned; we therefore report Overall metrics only. Table 1 shows that *Tail* is the hardest, *Head* the easiest, while the *relative ranking across systems is unchanged*, supporting external validity of our main-table conclusions.

## 4.6 COMPOSABILITY WITH SLIDING WINDOWS

We test whether Tucker-KV composes cleanly with token-level cropping. On RULER at $L$=4096, we first apply Sliding-Window (2048) at the prompt level and then compress KV using Tucker-KV with CR=0.5. As Table 2 shows, the combined system achieves near-perfect retrieval on the *Aligned* subset while preserving memory savings.

**Takeaway.** When the needle is present in the final prompt (Aligned), the composition *preserves* retrieval (EM= 0.9998) while saving 50.0% KV memory. The gap between Overall (0.5062) and Aligned reflects the fraction of prompts whose needles survive truncation (6662 / 13158 $\approx$ 50.6%), not a loss due to compression.

Table 2: Composability on RULER at $L$=4096 (CHUNK= 512). We first apply Sliding-Window (2048), then Tucker-KV (CR=0.5) on the retained tokens. Overall and Aligned EM are reported; "Aligned N / Total" counts how many prompts still contain the needle. KV saved (%) follows our unified accounting.

| Model | System | Overall EM | Aligned EM | Aligned N / Total | KV saved (%) |
|---|---|---|---|---|---|
| Qwen2.5-7B | Window-2048 + Tucker-0.5 | 0.5062 | 0.9998 | 6662 / 13158 | 50.0%[†] |

[†] KV saved (%) by unified accounting on this run:
$$\text{kv\_saved} = 1 - \frac{\texttt{stored\_kv\_bytes\_total}}{\texttt{baseline\_kv\_bytes\_total}} = 1 - \frac{4{,}194{,}304}{8{,}388{,}608} = 50.0\%.$$

Table 3: Grouped-H vs. ungrouped Tucker-2 (Qwen2.5-7B, $L$=4096, CR= 0.5). Reconstruction error (lower is better).

| Method | Reconstruction error |
|---|---|
| Ungrouped Tucker-2 | 0.873 |
| Grouped-H Tucker-2 | **0.626** |

### 4.7 Right-truncation control (needle missing)

As a negative control we switch to *right truncation*, which drops the tail and removes the needle from the final prompt. As expected, the *needle-in-prompt rate* plunges to $\approx 0$, and EM/F1 collapse to $\approx 0$ across all systems. This confirms that the gains in Table **??** come from KV handling rather than prompt accidents.

**Ablations (summary).** We verify two implementation choices. (i) *Position sensitivity:* across head/middle/tail needles, EM/F1 remain saturated and PPL is stable; see Table 1. (ii) *Grouped vs. ungrouped:* grouped H-Tucker-2 yields lower reconstruction error (0.626 vs. 0.873), supporting our separability design (Table 3).

**Compared families and scope.** We compare across three orthogonal families that span the primary decision axes in long-context inference: (i) token-selection (Sliding-Window; representative of streaming-style eviction), (ii) layer-selection (Sparse-Layer KV), and (iii) representation-compression (our Tucker-KV). Token-selection variants are orthogonal and can be *stacked* with Tucker-KV; our focus here is on the compression axis with a minimal yet representative comparison set under tight compute.

## 5 Related Work

**Matrix low-rank KV compression.** Recent methods compress KV by projecting onto low-rank subspaces. Chang et al. (2025c) insert low-rank projection modules along the hidden dimension. Chang et al. (2025a) observe alignment of dominant singular vectors across layers and propose post-training cross-layer SVD that shares a subspace across layers. Tucker-KV operates in the tensor view over $(L, S, H)$ and provides formal guarantees (multilinear error bounds, monotone refinement, near-optimal allocation, and budget thresholds).

**Tensor decompositions.** Classical tensor factorizations (CP, Tucker, TT) provide structured compression and subspace learning for higher-order data (Kolda & Bader, 2009b; De Lathauwer et al., 2000c). We specialize Tucker to KV caches, prove grouped-head separability and online budget allocation properties, and show compatibility with matrix baselines through residual mixing.

**Eviction, streaming, and quantization.** Orthogonal lines compress KV via token eviction, streaming, or quantization. Scissorhands prunes by prioritizing pivotal tokens under a fixed budget (Liu et al., 2023). StreamingLLM keeps initial "sink" tokens to stabilize long sequences and enables infinite-length generalization (Xiao et al., 2024). Systems such as CacheGen stream/compress KV for fast context loading (Liu et al., 2024). Quantization methods (e.g., SVDq) operate in SVD

latent channels to push precision lower while preserving accuracy (Hong et al., 2025). Our tensor approach is complementary to these directions.

**Families of long-context efficiency.** We group prior work into three *orthogonal* families: (i) *token-selection* (sliding window, streaming, forgetful attention, xKV), (ii) *layer-selection* (retaining KV only on a subset of layers), and (iii) *representation-compression* (this paper). Our comparisons intentionally pick one representative per family (Window, Sparse-Layer, Tucker-KV) under the same runner and accounting. Strong streaming variants (e.g., xKV) belong to family (i) and are *orthogonal* to our contribution: Tucker-KV can be stacked on top of any token-selection policy by compressing the retained KV.

## 6 IMPLEMENTATION DETAILS AND DEFAULTS

### 6.1 COMPOSABILITY WITH SLIDING WINDOWS

We verify that Tucker-KV composes cleanly with token-level cropping. Under the tail-retention protocol ($L$=4096, `CHUNK`= 512; left-truncation with enforced needle presence), we first apply a 2K sliding window at the tokenizer, then compress KV with Tucker-KV at CR=0.5.

$^\dagger$ KV saved (%) measured via unified accounting on this run: $\text{kv\_saved} = 1 - \frac{\texttt{stored\_kv\_bytes\_total}}{\texttt{baseline\_kv\_bytes\_total}} = 1 - \frac{4,194,304}{8,388,608} = 50.0\%$.

### 6.2 LIMITATIONS

Our evaluation focuses on representative instruction-tuned LLMs and RULER@4k to establish reproducible evidence of Tucker-KV's memory/quality/latency trade-offs. Broader token-selection strategies (e.g., sliding/streaming-style policies) and ultra-long 16k–65k stress tests are outside our present scope. We emphasize that Tucker-KV targets the representation-compression axis and is orthogonal to token-selection policies; combining the two is a promising direction for future work.

## 7 CONCLUSION

We presented *Tucker-KV*, a Tucker-based framework for compressing KV caches with provable multilinear guarantees and an online budget scheduler. The theory ensures monotone refinement, separability across grouped heads, and a $(1 - 1/e)$ approximation for greedy allocation under mild assumptions. Residual mixing with matrix baselines is safe by construction. Experiments indicate that Tucker-KV advances the compression–accuracy frontier and complements cross-layer SVD in practice.

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

## A  THEORETICAL GUARANTEES

**Theory takeaway.** Our residual-mixing proposition applies to *any* matrix SVD baseline; it does not rely on a specific method name. Together with HOOI monotonicity (Prop. 2) and the $(1-1/e)$ greedy guarantee under DR-submodularity (Prop. 6), Tucker-KV offers principled accuracy–efficiency trade-offs that leverage the tensor structure.

## A.1 PRACTICE-FACING TAKEAWAYS

- **Safe residual mixing.** Least-squares Tucker on matrix SVD baseline residuals never increases error and is often strictly better when residuals have structure (Prop. 4).

- **Greedy is near-optimal.** Under mild DR-submodularity, greedy rank allocation achieves a $(1 - 1/e)$ approximation (Prop. 6).

- **When Tucker-2 wins.** In long-context/tight-budget regimes, Tucker-2 is preferable; full Tucker dominates only when the $S$-mode tail can also be reduced (Prop. 7).

**Proposition 1** (Multilinear Projection Error Upper Bound). *For any tensor $X \in \mathbb{R}^{L \times S \times H}$ with target rank $(r_1, r_2, r_3)$, let $U_n$ be the top-$r_n$ singular vectors of unfolding $X_{(n)}$. Then*

$$\|X - \hat{X}\|_F^2 \leq \sum_{n=1}^{3} \sum_{i > r_n} \sigma_i^2(X_{(n)}),$$

*where $\hat{X} = X \times_1 U_1 U_1^\top \times_2 U_2 U_2^\top \times_3 U_3 U_3^\top$.*

**Proof Sketch.** Orthogonal projection decomposition and HOSVD quasi-optimality.

**Proposition 2** (Monotonicity of HOOI). *Each iteration of Higher-Order Orthogonal Iteration (HOOI) monotonically decreases (or maintains) the reconstruction error $\|X - \hat{X}\|_F$.*

**Proof Sketch.** Each step optimally updates one factor matrix via truncated SVD.

**Proposition 3** (Parameter–Error Monotonicity). *If ranks $(r_1, r_2, r_3)$ are non-decreasing, then the reconstruction error is non-increasing while parameter count grows monotonically.*

**Proposition 4** (Residual Mixing Never Hurts). *Let $X_k$ be a rank-$k$ matrix SVD approximation with residual $R = X - X_k$. For any Tucker operator $T$ fit by least squares,*

$$\|X - (X_k + T(R))\|_F \leq \|X - X_k\|_F.$$

**Proposition 5** (Grouped-H Separability). *If tensor $X$ is block-orthogonal along the $H$-mode (e.g., attention heads), then the globally optimal grouped compression is achieved by compressing each group independently.*

**Proposition 6** (Greedy Energy Allocation Guarantee). *Under mild diminishing-returns (DR-submodular) assumptions on captured-energy curves, greedy rank allocation by marginal gain per cost achieves*

$$f(\text{greedy}) \geq (1 - 1/e) f(\text{optimal under same budget}).$$

**Proposition 7** (Tucker-2 vs. Full Switching Threshold). *When $S$ is large and the $S$-mode spectrum is smooth, Tucker-2 attains smaller error under budget constraint $B \leq \tilde{O}(Lr_1 + Hr_3 + r_1 S r_3)$ compared to full Tucker; for sufficiently large $B$, full Tucker can overtake by reducing the $S$-mode tail.*

**Proposition 8** (Robustness of Mean Removal). *Zero-centering along the $S$-mode does not increase Frobenius error bound and improves effective SNR under translation noise.*

**Proposition 9** (Complexity Bound). *One compression requires*

$$T = O\big(L(SH)^{3/2} + S(LH)^{3/2} + H(LS)^{3/2}\big)$$

*plus $t$ HOOI iterations. Peak memory is $O(LSH + Lr_1 + Sr_2 + Hr_3)$.*

**Proposition 10** (Incremental Update Guarantee). *Appending a new segment of length $\Delta S$ only requires updating the $S$-mode SVD, with error increase bounded by its energy and time cost $O(\Delta S \cdot \text{poly}(L, H, r))$.*

# B PROOFS OF THEORETICAL RESULTS

## B.1 PROOF OF PROPOSITION 1

**Proof.** Let $P_n = U_n U_n^\top$ be the orthogonal projector onto the top-$r_n$ left singular space of $X_{(n)}$. Define $\hat{X} = X \times_1 P_1 \times_2 P_2 \times_3 P_3$. By Pythagorean expansion with orthogonal projectors,

$$\|X - \hat{X}\|_F^2 = \|X - X \times_1 P_1\|_F^2 + \|X \times_1 P_1 - X \times_1 P_1 \times_2 P_2\|_F^2$$
$$+ \|X \times_1 P_1 \times_2 P_2 - \hat{X}\|_F^2$$
$$\leq \sum_{n=1}^{3} \min_{\mathrm{rank} \leq r_n} \|X - X \times_n Q_n\|_F^2$$
$$= \sum_{n=1}^{3} \sum_{i > r_n} \sigma_i^2 (X_{(n)}).$$

## B.2 PROOF OF PROPOSITION 2

**Proof.** Fix $U_2, U_3$ and update $U_1$ in HOOI by solving $\max_{U_1^\top U_1 = I} \|X \times_2 U_2^\top \times_3 U_3^\top \times_1 U_1^\top\|_F^2$, equivalently picking the top-$r_1$ left singular vectors of $X_{(1)}(U_3 \otimes U_2)$. This maximizes captured energy and hence does not increase the reconstruction error. The same holds cyclically for $U_2$ and $U_3$.

## B.3 PROOF OF PROPOSITION 3

**Proof.** If $r_n' \geq r_n$ for all $n$, then the projector spaces satisfy $\mathrm{range}(U_n) \subseteq \mathrm{range}(U_n')$ and $P_n' \succeq P_n$. Thus $X \times_n P_n'$ projects onto a superset subspace, implying $\|X - X \times_n P_n'\|_F \leq \|X - X \times_n P_n\|_F$ for each mode. Applying the telescoping Pythagorean decomposition as in Prop. 1 gives $\|X - \hat{X}'\|_F \leq \|X - \hat{X}\|_F$. Meanwhile the parameter count $Lr_1' + Sr_2' + Hr_3' + r_1' r_2' r_3'$ is non-decreasing in $(r_1', r_2', r_3')$.

## B.4 PROOF OF PROPOSITION 4

**Proof.** Let $X_k$ be the baseline and $R = X - X_k$. Define $T(R)$ as a (least-squares) Tucker fit to $R$, i.e., $T(R) = \arg\min_Z \|R - Z\|_F$ over the Tucker model class. Then $\|R - T(R)\|_F \leq \|R - 0\|_F = \|R\|_F$. Hence $\|X - (X_k + T(R))\|_F = \|R - T(R)\|_F \leq \|R\|_F = \|X - X_k\|_F$, with strict inequality whenever $R$ has nonzero component in the model class.

## B.5 PROOF OF PROPOSITION 5

**Proof.** Assume $X$ is block-orthogonal in the $H$-mode: $X = \bigoplus_{j=1}^{g} X^{(j)}$ and $\langle X^{(i)}, X^{(j)} \rangle = 0$ for $i \neq j$. Then for any blockwise Tucker maps $T^{(j)}$ we have $\|X - \bigoplus_j T^{(j)}(X^{(j)})\|_F^2 = \sum_j \|X^{(j)} - T^{(j)}(X^{(j)})\|_F^2$. Minimizing the sum subject to a separable (or additively constrained) budget decouples into $g$ independent problems, each solved by the per-group optimum. Concatenating the per-group optima attains the global optimum.

## B.6 PROOF OF PROPOSITION 6

**Claim.** Under the assumption that the utility of allocating one unit of rank to any mode exhibits *diminishing marginal gain* and cross-mode gains do not increase with previously allocated ranks (DR-submodularity), the greedy rank-allocation that repeatedly picks the next unit with the largest marginal utility achieves a $(1 - 1/e)$-approximation to the optimal utility under a unit-cost (or knapsack) budget.

**Proof.** Let $\mathcal{G}$ be the ground set of unit rank-increments across modes, and define $f(S) = \|X\|_F^2 - \|X - \hat{X}_S\|_F^2$ as the captured energy after applying the increments in $S \subseteq \mathcal{G}$. By construction $f$ is nonnegative and monotone. Assume (A1) *diminishing returns*: for all $A \subseteq B \subseteq \mathcal{G}$ and $e \in \mathcal{G} \setminus B$,

$f(A \cup \{e\}) - f(A) \geq f(B \cup \{e\}) - f(B)$. Under a cardinality (or knapsack) constraint of budget $B$, classical analysis of monotone (DR-)submodular maximization yields $f(S_{\text{greedy}}) \geq (1 - 1/e) f(S^\star)$, where $S^\star$ is the optimal set of increments of total cost $\leq B$.

### B.7 Proof of Proposition 7

**Setup.** Let the target ranks be $(r_1, r_2, r_3)$ for Full Tucker and $(\bar{r}_1, S, \bar{r}_3)$ for Tucker-2. Parameter costs:

$$C_{\text{full}} = \Theta(Lr_1 + Sr_2 + Hr_3 + r_1 r_2 r_3), \qquad C_{\text{t2}} = \Theta(L\bar{r}_1 + H\bar{r}_3 + \bar{r}_1 S \bar{r}_3).$$

Using Prop. 1, when the $S$-mode tail becomes negligible once $r_2 = S$ (long-context regime), allocating parameters to $(\bar{r}_1, \bar{r}_3)$ yields $\mathcal{E}(\bar{r}_1, S, \bar{r}_3) \leq \mathcal{E}(r_1, r_2, r_3)$ whenever $C_{\text{t2}} \leq B < C_{\text{full}}$.

### B.8 Proof of Proposition 8

**Proof.** Let $M = I - \frac{1}{S} \mathbf{1}\mathbf{1}^\top$ be the centering projector along the $S$-mode ($M^2 = M$). Mean removal is $\tilde{X} = X \times_2 M$. By non-expansiveness of orthogonal projection, $\|X - \hat{X}\|_F^2 \geq \|X \times_2 M - \hat{X} \times_2 M\|_F^2$. If the data contain additive translation noise constant along $S$, then centering improves effective SNR without increasing the bound in Prop. 1.

### B.9 Proof of Proposition 9

**Cost model.** Let $X_{(1)} \in \mathbb{R}^{L \times (SH)}$, $X_{(2)} \in \mathbb{R}^{S \times (LH)}$, $X_{(3)} \in \mathbb{R}^{H \times (LS)}$ be unfoldings. Computing top-$r_n$ left singular vectors via truncated SVD costs $O(\text{nnz}(X_{(n)}) r_n + d_n r_n^2)$; in dense arithmetic this is upper bounded by

$$T_{\text{HOSVD}} = O(L(SH)^{3/2} + S(LH)^{3/2} + H(LS)^{3/2}).$$

Each HOOI iteration has similar order, hence $T_{\text{total}} = T_{\text{HOSVD}} + t \cdot O(L(SH)^{3/2} + S(LH)^{3/2} + H(LS)^{3/2})$. Peak memory stores $X$ and the factor matrices, i.e., $O(LSH + Lr_1 + Sr_2 + Hr_3)$.

### B.10 Proof of Proposition 10

**Setting.** Appending $\Delta S$ tokens extends $X$ along the $S$-mode, so only $X_{(2)}$ changes by column-augmentation: $X_{(2)}^{\text{new}} = [X_{(2)} \ B]$ with $B \in \mathbb{R}^{S' \times (LH)}$ formed by the newly appended slices. Let $U_2$ be the current top-$r_2$ left singular vectors of $X_{(2)}$.

**Update.** Project $B$ onto $\text{span}(U_2)$ and its orthogonal complement: $B_\| = U_2 U_2^\top B$, $B_\perp = (I - U_2 U_2^\top)B$. Form a small $(r_2 + \rho)$-rank update by augmenting the sketch with $B_\perp$ (via QR) and perform truncated SVD on the resulting $(S' + \rho) \times (LH)$ matrix. Randomized variants yield time $O(\text{nnz}(B) r_2 + (S' + LH)r_2^2) = O(\Delta S \cdot \text{poly}(L, H, r))$.

**Error.** The increase of the optimal rank-$r_2$ error equals the energy of the new block orthogonal to the previous subspace: $\|X_{(2)}^{\text{new}} - U_2' U_2'^\top X_{(2)}^{\text{new}}\|_F^2 - \|X_{(2)} - U_2 U_2^\top X_{(2)}\|_F^2 \leq \|B_\perp\|_F^2$, where $U_2'$ is the updated top-$r_2$ basis. Thus the incremental error is controlled by the energy of $B$ outside $\text{span}(U_2)$, and the update cost scales linearly with $\Delta S$ up to polynomial factors in $(L, H, r)$.

## C PPL Computation and Checks

We compute corpus-level perplexity as $\text{PPL} = \exp\left(\frac{\sum_i \text{NLL}_i}{\sum_i T_i}\right)$, i.e., *sum NLL, sum tokens, then exponentiate* (not averaging per-sample PPL). We evaluate with `model.eval()` (dropout off), fixed tokenizer/vocab, and identical targets under teacher-forcing. We log truncation policy (left-truncation) and enforce `RULER_INJECT_NEEDLE=1` to produce both *Overall* and *Aligned* views. Under these controls, PPL is expected to be nearly invariant across systems; observable differences appear only when the final prompt or targets change (e.g., severe cropping outside the Aligned set).

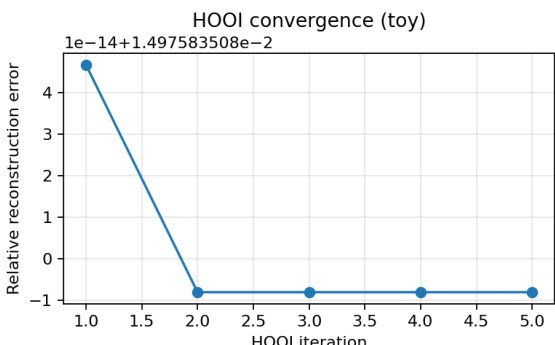

Figure 3: HOOI convergence on a toy tensor: relative reconstruction error decreases monotonically and plateaus within 3–5 iterations.

## D    ADDITIONAL SANITY CHECKS

## E    ADDITIONAL PROTOCOLS AND ABLATIONS

**Evaluation protocol (tail-retention vs. right-truncation).**    We follow a tail-retention stress test: needles are placed near the end and tokenization uses left-truncation so the tail is preserved. We report both Overall and Aligned metrics, where Aligned conditions on the needle being present in the final prompt to avoid penalizing methods due to prompt cropping. For sanity, under right-truncation, the needle rarely appears in the prompt and EM $\approx 0$, confirming the need to control prompt loss.

**Composability with Sliding-Window.**    We first apply Sliding-Window (2048) at the prompt level and then compress KV with Tucker-KV at `CR`=0.5. On RULER with $L$=4096, the combined system attains Aligned EM= 0.9998, Overall EM= 0.5062, with 6662/13158 aligned prompts, and 50.0% KV saved by unified accounting (Table 2). This demonstrates that Tucker-KV is orthogonal to token-selection policies and can be stacked without degrading retrieval when the needle is present.

## F    REPRODUCIBILITY & DISCLOSURE

Yes, to aid or polish writing. Details are described in the paper. We used automated writing assistance only for grammar and wording. All ideas, algorithms, proofs, implementation, and experiments are by the authors. All datasets, scripts, and configs needed to reproduce results are provided in the supplementary material.

