# OpenReview forum: "Tucker-KV: Provable Tucker Compression of KV Caches with Monotone Refinement and Near-Optimal Budgeting"
_ICLR.cc/2026/Conference — ICLR 2026 Conference Desk Rejected Submission_

### Official Review · Reviewer_HxxC · 2025-10-30

**Soundness:** 2
**Presentation:** 1
**Contribution:** 3
**Rating:** 4
**Confidence:** 3

**Summary:**

This paper analyzes KV Cache from the tensor perspective, where $X \in \\mathbb{R}^{L \times S \times H}$. The authors use Tucker compression technique to approximate the original $X$ to reduce the memory size of the kv cache, and they have proven the error convergence (Proposition 1). Another contribution is the authors prove that each HOOI iteration never increases the reconstruction error (Proposition 2), which gives a monotone refinement guarantee for support their Tucker approximation convergence.

**Strengths:**

$\bullet$ Tensor version of KV Cache analysis: This paper addresses the KV cache compression from the tensor perspective, and the authors have proven the multi-linear error control and monotone reconstruction error across iterations. Tensor version gives us more information and more complicated analysis in comparison with the matrix version.

$\bullet$ Flexible Design: works per head-group can be stacked on top of a strong matrix-SVD baseline via residual mixing which won't increase error.

$\bullet$ Good performance: TuckerKV achieves large KV memory reductions and keeps task quality on long-context evaluations

**Weaknesses:**

$\bullet$ Unrealistic Assumption: In Proposition 5, the authors seems to derive the global optimality based on tensor $X$ is block-orthogonal along the $H$-mode (eg. attention heads) on Line 574. In practice, I think heads are not orthogonal empirically. In [1], it states '' Our novel pruning methods removes the vast majority of heads without seriously affecting performance.'' Moreover, In [2], it states " most attention heads can be individually removed after training without any significant downside in terms of test performance". If heads group were orthogonal, you wouldn't be able to to prune most of them with a small loss. Therefore, I think the condition in your Proposition 5 is unrealistic.

$\bullet$ Poor presentation and organization: This paper is structured differently from most publications at top ML conferences (NeurIPS, ICLR, ICML, etc.). The introduction is broken into too many sections (“This paper”/“Contributions”/“Scope”/“Design rationale”), and the structure is not logical, which makes it harder for the reader to understand what the authors are trying to say. For instance, a better structure could follow: Background → Problem → Gap → Approach → Contributions → Evidence and Implications. The authors could also formulate a clear research question so readers can quickly understand the problem they are trying to solve.

As far as I understand this paper, one of the main contributions is theoretically proving the convergence of the tensor-version KV-cache Tucker compression, and the empirical results validate the theoretical findings. I think it will be presented better if author can include a main theorem or proposition in the main body rather than putting all theoretical findings in the appendix. The Related Work section should follow the Introduction, and a preliminaries section providing background information typically exists. it is not mandatory to structure a paper like this, but the authors should structure the paper in a logical way so readers can easily follow. I strongly recommend the authors check other publications in related work to improve the overall presentation of this paper.


[1] Voita, Elena, et al. "Analyzing multi-head self-attention: Specialized heads do the heavy lifting, the rest can be pruned." ACL'19.

[2] Michel, Paul, Omer Levy, and Graham Neubig. "Are sixteen heads really better than one?" NeurIPS'19.

**Questions:**

$\bullet$ Can author explain why the assumption in Proposition 5 is realistic?

$\bullet$ Does the main result still hold when the assumption in Proposition 5 is removed?

---

### Official Review · Reviewer_wJa2 · 2025-10-30

**Soundness:** 2
**Presentation:** 1
**Contribution:** 2
**Rating:** 2
**Confidence:** 2

**Summary:**

The paper presents Tucker-KV, a method for compressing the key–value caches in Transformers using a Tucker tensor decomposition. Experiments on Qwen2.5-7B with the RULER benchmark report roughly 83% KV memory savings without loss in task accuracy or perplexity, suggesting the method could be an effective drop-in compression scheme.

**Strengths:**

The idea of using Tucker decomposition for KV-cache compression seems novel, compared to previous approaches that only considers low-rank decomposition using SVD (and not high-order SVD for Tucker decomposition). The approach appears compatible with other KV efficiency strategies such as windowed or sparse strategies.

**Weaknesses:**

Overall I feel that the paper could gain in clarity by further reinforcing the narrative and the motivation at different stages of the paper. Concretely:

* The paper could benefit from greater clarity by properly introducing KV caches at a high level. In particular, it does not motivate why KV compression should work: the reader may wonder what structural properties of KV tensors make them amenable to compression, or whether a low-rank structure is theoretically justified or empirically observed. Without this context, it is difficult to understand why Tucker decomposition is an appropriate modeling choice rather than an arbitrary compression technique.
* Section 2.2 on online budget allocation is difficult to follow and appears insufficiently motivated at this point in the paper. It is introduced abruptly, before the reader fully understands the problem setting or why such an allocation mechanism is needed. A brief intuitive explanation or contextual motivation earlier on would make this section much easier to grasp.
* The contribution paragraph in the introduction is currently a list rather than a cohesive narrative. I suggest the authors rework it to better articulate and emphasize the paper’s key novelties.
* I feel that some technical steps are underspecified, e.g., it is unclear whether the method uses a full Tucker or Tucker-2 decomposition to do KV-cache compression.
* In section 2.7, it is not clear how the truncated SVD to compute $X_k$ is performed (on which flattened dimensions), or why applying Tucker to the residual $R = X - X_k$ is expected to work.
* The authors could make the description of the experimental protocol more self-contained by describing a little more the RULER benchmark, along with the concept of "needle", and why this benchmark is challenging and relevant to study KV cache compression. Otherwise, the reader might be confused about the specific choice of this dataset to validate the method.
* The set of baselines in the experimental protocol feels incomplete. While comparisons to sliding-window and sparse methods are informative, they are not the most relevant since, as the authors themselves note, these approaches are orthogonal to Tucker-KV. A more appropriate evaluation would include baselines from the same family of KV-compression methods, such as standard low-rank or SVD-based approaches, which are currently missing from the submission. Without such a comparison, it is not clear whether the proposed approach is relevant in practice.
* Table 1 reports comparisons on Qwen2.5-7B, but for Llama 3.1 only the Tucker-KV results with a compression ratio of 1.0 (meaning 0% KV saved) are shown. It is unclear why results for other compression settings or baseline methods are missing, which makes the comparison appear incomplete.
* Overall the paper did not discuss the reconstruction error $ \|X - \hat{X}\| $ where $X$ is the full KV cache encountered in practice during inference, and $\hat{X}$ is its compressed version. In particular, this leaves unclear how approximation quality relates to model performance during inference.
* All the theoretical results are presented only in the appendix and not discussed in the main text. This may leave readers uncertain about the relevance and role of these results within the overall contribution of the paper.
* The authors mention using HOSVD and HOOI to perform the Tucker decomposition, but it is not clearly stated whether this constitutes a new algorithmic contribution or simply the application of existing methods from the literature. The lack of explicit references makes this point ambiguous.

**Questions:**

1. What structural properties of KV tensors justify a low-rank tensor approximation?
2. When exactly is the KV cache compressed during decoding, and how is it updated afterward?
3. How do you choose between Tucker and Tucker-2 in practice?
4. On which dimensions is the cache flattened when performing the SVD for \(X_k\) in Algorithm 2?
5. Why should applying Tucker to the residual guarantee improvement?
6. Why are there no comparisons with other KV-compression methods (e.g., SVD or CP decomposition)?
7. Could you report the reconstruction error and discuss its link to downstream metrics?

---

### Official Review · Reviewer_irCz · 2025-11-01

**Soundness:** 2
**Presentation:** 2
**Contribution:** 3
**Rating:** 2
**Confidence:** 5

**Summary:**

The paper proposes Tucker-KV, a method for compressing transformer KV caches using Tucker tensor decomposition. Unlike prior works that flatten the cache into a matrix and apply SVD, Tucker-KV treats the cache as a 3D tensor (hidden × sequence × head) and compresses along all modes. The authors provide comprehensive theoretical analysis to justify Tucker’s use in KV compression, offering provable guarantees such as monotone refinement and error bounds. While the theoretical exposition is strong, the experimental scope is limited. Experiments on Qwen2.5-7B and Llama-3.1-8B show that Tucker-KV maintains full model quality while saving up to 83% of KV memory, but only under short-context RULER tasks.

**Strengths:**

**Originality:** Applying Tucker decomposition to explore the low-rank structure of KV caches across multi-head and multi-layer architectures is an insightful and novel direction.

**Theoretical foundation:** The paper provides solid theoretical grounding, showing Tucker’s better reconstruction error bound compared to matrix SVD and offering proofs for refinement and safety properties.

**Weaknesses:**

Despite the promising idea, several critical evaluation and design aspects are missing:

+ **No runtime or cost analysis:** The paper does not measure the actual computational or latency cost of Tucker compression. There are no wall-clock timings, FLOPs, or GPU utilization reports for prefill or decoding, making it unclear how expensive Tucker-KV is in practice.

+ **No direct comparison with SVD-only baseline:** Although Tucker-KV includes a residual SVD path and proves that the combination is guaranteed not to degrade accuracy, the paper provides no experimental results directly comparing SVD-only, Tucker-only, and hybrid variants at equivalent compression ratios. Without this, it is difficult to assess Tucker’s practical advantage over well-established SVD methods.

+ **Limited evaluation scope:** The experiments are confined to 4k-context RULER benchmarks. At such short contexts, KV caches contribute only a small portion of total memory, meaning compression is not yet critical. The paper explicitly excludes long-context stress tests (e.g., 16k–128k), where KV compression truly matters, labeling them as out of scope. This omission weakens the empirical significance of the work.

+ **Lack of inference integration details:** The paper does not explain how Tucker-compressed KV caches are used during real inference. For example, does Tucker-KV only apply to prefill caches? Are the decomposed tensors reconstructed for every decoding step? How are the decomposed factors accessed in attention computation? These are crucial practical questions for determining whether Tucker-KV can actually save time or memory in live inference.

**Questions:**

1. Can you provide runtime and latency analysis for Tucker-KV, including prefill and decoding stages, to quantify the overhead introduced by Tucker decomposition compared to standard SVD and Full-KV? It might be good to test it with a different context size.

2. How exactly is the Tucker-compressed cache integrated into the inference process? Is the KV tensor reconstructed before each attention operation, or are the decomposed factors used directly during decoding?

3. Can you include a direct comparison among SVD-only, Tucker-only, and the proposed hybrid residual design at the same compression ratio, to demonstrate whether Tucker provides measurable practical gains?

---

### Official Review · Reviewer_atDj · 2025-11-01

**Soundness:** 1
**Presentation:** 1
**Contribution:** 2
**Rating:** 2
**Confidence:** 3

**Summary:**

The paper proposes a KV cache compression framework based on Higher-Order SVD. It provides theoretical guarantees for this compression method and empirically demonstrates that it achieves a favorable trade-off on RULER@4k using the Qwen2.5-7B model.

**Strengths:**

- Provides a theoretical error bound for the KV cache compression method.
- Proposes a practical online budgeting mechanism.

**Weaknesses:**

**[W1]** Limited empirical validation: The experiments are primarily conducted on the Qwen2.5-7B model. Furthermore, only very basic KV cache compression methods are used as baselines. Although the paper states that the focus is on the representation-compression axis, there exist much more advanced KV cache representation compression methods that should be compared against, such as xKV (https://arxiv.org/abs/2503.18893), KIVI (https://arxiv.org/pdf/2402.02750), and GEAR (https://arxiv.org/abs/2403.05527).

**[W2]** The paper is difficult to follow:
- It is not immediately clear which dimensions (L, S, H) refer to which component from the abstract without further description.
- The font size in Figure 1 is too small, making it difficult to read.

**Questions:**

**[Q1]** L102: What does the success flag represent? The success of which process?

**[Q2]** How would the method perform on reasoning models such as the Qwen3 model family for long-generation tasks like AIME or LiveCodeBench? Demonstrating this would highlight the method’s robustness under long-generation scenarios, which are not captured by long-context retrieval tasks.

---

### Note · Program_Chairs · 2026-01-17
**Submission Desk Rejected by Program Chairs**

The following references in this submission do not refer to real documents and/or have major errors in bibliographic information:

 Yuxin Han et al. Kv cache compression in llm inference: A survey. arXiv:2403.05527, 2024. https://arxiv.org/abs/2403.05527.